# Cactus Pear as Roughage Source Feeding Confined Lambs: Performance, Carcass Characteristics, and Economic Analysis

Kleitiane Balduíno da Silva [1], Juliana Silva de Oliveira [1], Edson Mauro Santos [1], João Paulo de Farias Ramos [1], Felipe Queiroga Cartaxo [2], Patrícia Emília Naves Givisiez [1], Aelson Fernandes do Nascimento Souza [1], Gabriel Ferreira de Lima Cruz [1], José Maria César Neto [1], Joyce Pereira Alves [1], Daniele de Jesus Ferreira [3], Anny Graycy Vasconcelos de Oliveira Lima [3] and Anderson de Moura Zanine [3,*]

1. Department of Animal Science, Federal University of Paraíba, Areia 58397-000, Paraíba, Brazil; kleitezootec@gmail.com (K.B.d.S.); oliveirajs@yahoo.com.br (J.S.d.O.); edsonzootecnista@yahoo.com.br (E.M.S.); jpemepapb@yahoo.com.br (J.P.d.F.R.); patriciagivisiez@yahoo.com.br (P.E.N.G.); aelsonzoo@gmail.com (A.F.d.N.S.); g_ferreira_dm@hotmail.com (G.F.d.L.C.); netocesar2511@hotmail.com (J.M.C.N.); joycepereira_alves@hotmail.com (J.P.A.)
2. Center for Human and Agricultural Sciences, State University of Paraíba (UEPB), Campina Grande 58429-500, Brazil; felipeqcartaxo@yahoo.com.br
3. Department of Animal Science, Federal University of Maranhão, Chapadinha, Maranhão 65500-000, Brazil; daniele.ferreira@ufma.br (D.d.J.F.); annygraycy@gmail.com (A.G.V.d.O.L.)
* Correspondence: anderson.zanine@ufma.br

**Abstract:** The effect of diets containing 0% of wheat bran levels (control buffel grass and cactus pear) contrasted with diets with cactus pear as the only forage source and wheat bran levels (30; 37 and 44%) on nutrient digestibility, feed intake, animal performance, carcass characteristics, and economic analysis was evaluated. Twenty-eight male, non-castrated crossbred lambs (22.6 ± 2.37 kg) were submitted to confinement for 62 days. A completely randomized design was used with four treatments and seven repetitions. Four diets were formulated considering an intended mean daily weight gain of 200 g/animal/day. Means were compared by contrasts using Dunnett test at 5%. Animals fed cactus pear as the only roughage source (diets with 30; 37, and 44% of wheat bran) had lower dry matter intake and nutrient intake, Average Dairy Gain, and total weight gain than animals fed the control diet. Final body weight and slaughter weight of animals fed 44% of wheat bran was similar to the animals fed the control diet. Hot and cold carcass yields were higher in animals fed cactus pear and 30 and 37% of wheat bran. Feeding costs were lower when cactus pear was used as the only roughage source associated with wheat bran and consequently profit was greater. The use of cactus pear as the only roughage source associated with up to 44% of wheat bran is a viable alternative of the diet to confined lambs without modifying carcass characteristics with greater cost:benefit ratio.

**Keywords:** carcass weight; cactus pear cladodes; physically effective fiber; weight gain; wheat bran

## 1. Introduction

Livestock farming is an intensive practice in semiarid regions, such as some country in South America, Africa, and the Middle East, especially small ruminant farming (goats and sheep). The long draught periods in such regions jeopardize forage quality and availability, and consequently might result in poor animal performance. Therefore, confinement techniques can enable feeding animals with diets with higher nutritional value and prevent energy expenditure due to grazing and search for feed [1].

Cactus pear (*Opuntia ficus indica* or *Nopalea cochonillifera*) is a favorable option to be used as animal forage in semiarid regions, since it is adapted to the edaphoclimatic conditions and provides high forage production, contributing for the feed availability. Besides, it is used as source of water, non-fibrous carbohydrates, and energy [2]. On the

other hand, cactus pear cannot be used as the only fiber source, because it has limitations concerning the levels of dry matter (9.2%), crude protein (6.3%), and neutral detergent fiber (21.7%) on a dry matter basis [3], and does not meet ruminant nutritional requirements.

Considering such characteristics, when cactus pear is used as the only fiber source in ruminant feeding, the result might be liquid feces and weight loss [4,5]. It is thus recommended that cactus pear is given to animals together with a fiber source to prevent these digestive problems [5,6].

Wheat bran can contribute to minimize the nutritional limitations of cactus pear. As an energy concentrate feedstuff, it contains medium levels of neutral detergent fiber (35%) and high levels of crude protein (17.4%) [7]. Furthermore, it is highly available to producers, it is inexpensive since it is a by-product, and the price is lower than the cost of forage hay and silage during drought periods in semiarid regions. The difficulty to produce enough forage to meet fiber requirements of confined animals due to climatic limitations must also be considered. Thus, the association between cactus pear and wheat bran can be a promising alternative in sheep confinement, besides allowing this practice in farms that exclusively produce cactus pear [8].

The nutritional importance of the cactus pear in lambs finishing has been previously reported [9–11]. Although some studies have assessed the effect of cactus pear as the only forage source [12–14] they have not included any additional fiber source in the diet such as wheat bran. These are, however, incipient studies and have not provided many results concerning performance, digestibility, intake and carcass traits. Besides, previous studies have not included economic analysis, thus limiting the proof of efficacy of adopting cactus pear as the only forage source in finishing diets without decreasing the performance. Additionally, the use of cactus pear associated wheat bran can improving environmental sustainability in the production of ruminants [15].

This study assessed the nutrient intake, digestibility, performance, carcass characteristics, and economic analysis of diets containing cactus pear as the only forage source associated with different levels of wheat bran in comparison to a standard diet with cactus pear and buffel grass hay.

## 2. Materials and Methods

### 2.1. Location and Meteorological Data

The experiment was carried out in the confinement installations of the Experimental Farm Benjamin Maranhao, from Empresa Paraibana de Pesquisa, Extensão Rural e Regularização Fundiária (EMPAER), located in Tacima, PB, Brazil. The coordinates of the experimental farm are 35°38 W and 6°29 S with typical precipitation of 431.8 mm/year, *Bsh* Köppen' climate classification, and a temperature of approximately 26 °C between March and May of 2018.

### 2.2. Animals, Diet, and Management

The study used 28 male, non-castrated and undefined crossbred lambs with initial weight of approximately 22.6 kg ± 2.37 kg, and mean age of 150 ± 17 days old. The period for adaptation to diets and installations lasted ten days, and data collection lasted 52 days. The animals were identified, weighed, vaccinated against clostridiosis, and treated against ecto- and endoparasites before distribution in individual slatted pens measured 1.5m$^2$ with feeders and water supply. The lambs were subjected to the daily cleaning management of the stalls, as well as the removal of the feed refusals, cleaning of the feeders and water fountain.

A completely randomized experimental design with four treatments and seven repetitions was used. Treatments consisted of a standard control diet with buffel grass hay, concentrate and cactus pear cladodes, and three diets comprised of cactus pear as the forage and different concentrations of wheat bran on a dry matter basis (30%, 37%, or 44%).

Cladodes originated from plants cultivated at the Experimental Farm. Secondary, tertiary, quaternary, and further cladodes were harvested and given to the animals, whereas

mother and primary cladodes were preserved. Experimental diets contained approximately 2 cm chopped cactus pear cladodes (*Nopalea cochenillifera* Salm-Dyck), buffel grass hay (*Cenchus ciliaris*), soybean meal, corn, wheat bran, and mineral supplements (Tables 1 and 2).

**Table 1.** Chemical composition of the diet ingredients on a dry matter basis (DM).

| Ingredient, g kg$^{-1}$ DM | Cactus Pear | Buffel Grass Hay | Corn Meal | Soybean Meal | Wheat Bran | Urea |
|---|---|---|---|---|---|---|
| Dry matter [†] | 169.72 | 836.17 | 857.01 | 718.52 | 812.66 | 995.40 |
| Organic matter [‡] | 155 | 777 | 847.42 | 672.79 | 773.01 | 990.02 |
| Crude protein | 29.60 | 95.33 | 95.12 | 480.70 | 164.17 | 2810 |
| Ether extract | 21.21 | 18.58 | 50.28 | 34.13 | 45.54 | - |
| NDFcp [§] | 182.27 | 714.66 | 104.27 | 242.90 | 394.22 | - |
| ADF [¶] | 88.29 | 387.08 | 23.91 | 98.33 | 108.29 | - |
| Mineral matter | 81.43 | 70.53 | 11.57 | 64.20 | 47.98 | 4.60 |
| Total carbohydrates | 870.4 | 813.0 | 753.1 | 421.0 | 742.3 | - |
| Non-fiber carbohydrates | 664.1 | 544.3 | 604.1 | 148.2 | 331.2 | - |
| Cellulose | 77.81 | 287.79 | 20.54 | 94.44 | 88.34 | - |
| Hemicellulose | 93.06 | 327.51 | 80.18 | 144.46 | 286.80 | - |
| Lignin | 10.47 | 99.28 | 3.37 | 3.89 | 19.94 | - |

[†] On natural matter basis; [‡] on dry matter basis; [§] neutral detergent fiber corrected for ash and protein; [¶] acid detergent fiber.

**Table 2.** Percentage and chemical composition of the experimental diets on a dry matter basis.

| Item | Diets [†] | | | |
|---|---|---|---|---|
| | 0% WhB | 30% WhB | 37% WhB | 44% WhB |
| Ingredient (g kg$^{-1}$ DM) | | | | |
| Cactus pear | 382.64 | 382.73 | 382.9 | 382.69 |
| Buffel grass hay | 258.33 | 0 | 0 | 0 |
| Soybean meal | 154.04 | 71.64 | 56.94 | 42.19 |
| Corn meal | 184.16 | 232.58 | 177.7 | 114.02 |
| Wheat bran | 0 | 294.41 | 363.36 | 441.57 |
| Urea | 1.96 | 0 | 0.49 | 0.79 |
| Mineral supplement | 11.77 | 11.78 | 11.78 | 11.78 |
| Ammonium chloride | 6.87 | 6.87 | 6.87 | 6.87 |
| Ammonium sulphate | 0.22 | 0 | 0.05 | 0.09 |
| Chemical composition (g kg$^{-1}$ MS) | | | | |
| Dry matter [‡] | 282.05 | 259.41 | 269.5 | 255.86 |
| Organic matter [§] | 257.75 | 237.27 | 249.44 | 234.08 |
| Crude protein | 127.69 | 129.95 | 129.71 | 122.66 |
| Ether extract | 27.1 | 34.66 | 34.54 | 34.4 |
| Neutral detergent fiber | 343.48 | 253.3 | 269.45 | 288.07 |
| Physically effective fiber | 329.13 | 231.43 | 261.22 | 269.42 |
| Mineral matter | 88.25 | 83.72 | 85.56 | 82.92 |
| Total carbohydrates | 513.1 | 439.8 | 439.2 | 450.3 |
| Non-fiber carbohydrates | 268.21 | 317.3 | 322.92 | 329.63 |
| Metabolizable energy | 2.52 | 2.72 | 2.71 | 2.71 |
| Cellulose | 122.01 | 67.24 | 70.82 | 75.06 |
| Hemicellulose | 157.1 | 149.04 | 162.54 | 177.32 |

[†] 0% WhB was Control = buffel grass hay, cactus pear and concentrate; 30% WhB = cactus pear, 30% wheat bran on DM basis and concentrate; 37% WhB = cactus pear, 37% wheat bran on DM basis and concentrate; 44% WhB = cactus pear, 44% wheat bran on DM basis and concentrate. [‡] g kg$^{-1}$ on a natural matter basis. [§] g kg$^{-1}$ on dry matter basis.

Diets were calculated based on the requirements of confined lambs with an initial weight of 22 kg and daily weight gain of 200 g per animal per day [16].

Feed and water were supplied *ad libitum*. Diets were given twice daily (08:00 and 16:00 h), and voluntary feed intake was calculated in order to adjust the amount of offered

diet, considering 10% of additional feed. Nutrient estimate intake was calculated using the difference between the averages of total nutrients in the offered diet and the total nutrient amount in the feed refusals.

Voluntary water intake was calculated as the difference between the amount provided and remaining water in the buckets in 24 h. Intake was corrected for evaporation rate, which was monitored using similar buckets and water volume placed outside the pens.

On the first day (D1) and day 52 (D52) of the experimental period, animals were fasted for 16 h and weighed to determine initial body weight (IBW) and final body weight (FBW), respectively, and total weight gain (TWG) was determined. Weight gain was determined at every two weeks, and average daily gain (ADG) was obtained by TWG divided by the total confinement days. Feed conversion was obtained by the dry matter intake (g/day) divided by the ADG (g/day). Feed efficiency was calculated by the ADG (g/day) divided by the dry matter intake (g/day).

### 2.3. Preparation of Samples and Analysis

Ingredients were sampled before every preparation of the experimental diets, whereas offered feed and feed refusals in the feeder were sampled weekly. Fecal samples were collected for three consecutive days, pooled, and frozen (−15 °C) until pre-drying and analysis.

Bromatological analysis were carried out on the Animal Nutrition Laboratory from the Animal Science Department of the Federal University of Paraíba. The methods of Association of Official Analytical Chemists [17] were used to determine dry matter (DM, method 934.01), crude protein (CP, method 954.01), ether extract (EE, method 920.39), mineral matter (MM, method 942.05), and lignin (method 973.18). ANKOM 200 Fiber Analyzer (ANKOM Technology Corporation, Fairport, NY, USA) was used to determine the neutral detergent fiber (NDF) and acid detergent fiber (ADF) [18]. NDF was corrected for protein and ash, which was determined by incineration in a muffle at 600 °C for four hours. NDF was corrected for ash and protein (NDFap) according to Licitra et al. [19] and Mertens [20]. Hemicellulose was calculated as the difference between NDF and ADF.

Total carbohydrates were estimated using the equation TC = 100 − (%CP + %EE + %Ash) [21]. Non-fiber carbohydrates (NFC) were calculated with the equation proposed by Hall [22] for feed containing urea: NCF = 100 − [(%CP − (%CP urea + %urea)) + %NDFap + %EE + %Ash], in which %CP urea and NDFap indicate the crude protein from urea and neutral detergent fiber corrected for ash and protein, respectively.

Physically effective fiber (peNDF) was determined according to Kononoff et al. [23] by passing the experimental diets through a series of sieves with progressively smaller from top to bottom (19 mm, 8 mm, and 1.18 mm) and a lid on the bottom. The percentage of retained particles in each sieve (screen) was calculated as the weight of the fraction in each sieve divided by the total weight of all fractions. peNDF was estimated using the formula peNDF = NDFap × % total retained fraction.

### 2.4. Apparent Digestibility of Diets

The digestibility trial was carried out on day 15 (D15) of the experimental period, by sampling the diets, feed refusals, and feces. The latter was collected twice a day from the rectal of the animals for four consecutive days: D15 (06:00 h and 14:00 h), D16 (8:00 h and 16:00 h), D17 (10:00 h and 18:00 h), and D18 (12:00 h and 20:00 h). Feces samples were weighed, identified, and stored at −15 °C. On D18, all samples of one animal were pooled, homogenized, and one composite sample per animal was pre-dried in a forced circulation drying oven at 65 °C for 72 h.

All diets, feed refusals, and composite feces samples were ground using a knife mill with a two-mm sieve before analysis. FDMP (fecal DM production) was estimated using the indigestible neutral detergent fiber (iNDF) as an internal marker. iNDF concentrations were determined in separate using samples of concentrates (1g) in each bag and samples of hay, feces, and feed refusals (0.5g, in each bag) were incubated in a non-woven-fabric

bag inserted for 288 h in the rumen of a fistulated bovine [24]. The fistulated bovine was fed a 50:50 ratio roughage:concentrate diet. The remaining material after incubation was digested with neutral detergent and the residue was considered iNDF according to the method INCT-CA F/011/1 as described by Detmann et al. [25].

FDMP was determined using the formula FDMP = marker intake (kg)/marker concentration in feces (%). The digestibility coefficients of DM, OM, CP, EE, and NDF were calculated as CD = [(nutrient intake in grams − nutrient in feces in grams)/(nutrient intake in grams)] × 100. The digestibility coefficient of NFC was estimated from calculated the amount of NFC in the diets, feed refusal, and feces.

### 2.5. Slaughter and Carcass Evaluation

The final live body weight was determined at the end of the confinement period. The animals were fasted for 16 h and then weighed to determine the slaughter body weight (SBW) and the post-fast weight loss.

The slaughter was performed according to the Regulation of Industrial and Sanitary Inspection of Products of Animal Origin (RIISPOA) [26]. Briefly, the animals were stunned using a captive bolt gun and cerebral concussion, followed by a four-minute exsanguination after carotid artery and jugular vein severance. The blood was collected for weight determination. After dressing and evisceration, the head and the feet were removed by sectioning the neck joint, and the metacarpal and metatarsal joints, respectively. The hot carcass weight (HCW) and the weights of the thoracic, pelvic, and abdominal organs were determined. The carcasses were suspended by the gastrocnemius muscle tendon at 4 °C for 24 h before determining the cold carcass weight (CCW) [27].

The full and empty weights of the gastrointestinal tract (GIT) were determined to calculate the empty body weight (EBW) and true yield percentage (TYP). TYP (%) = HCW/EBW × 100. The kidneys and perirenal fat were removed and subtracted from the hot and cold carcass weights to determine the hot carcass yield (HCY = HCW/SBW × 100); cold carcass yield (CCY = CCW/SBW × 100); and the chilling loss [CL = (HCW − CCW)/HCW × 100], according to Cezar and Sousa [27].

The carcasses were split down the median plane to yield two half-carcasses that were refrigerated for 24 h at 4 °C. Viscera and organs were weighed (blood, liver, heart, kidneys, lungs, empty intestines, gall bladder, tongue, and spleen). Internal fat included the perirenal, pelvic, omental, and mesenteric fat. Empty body weight (EBW) was calculated as the difference between slaughter body weight and the gastrointestinal weight. The residues of the carcass were weighed (skin, head, feet, tail, internal fat, testicles, and blood).

After the chilling period, the carcasses were split down the median plane and the two half-carcasses were weighed. In the left half-carcass, the length (internal and external), leg length, thorax perimeter, width, and depth, and hind width and perimeter were measured according to Cezar and Sousa [27]. The half-carcasses were then sectioned into five regions of commercial cuts [27], as follows: Neck, shoulder, ribs, loin, and leg. Commercial cut yield was calculated by each individual weight as a percentage of the half-carcass.

A transversal cut was made between the 12th and 13th ribs to expose the transversal section of the *Longissimus dorsi* muscle. The loin eye area was measured using a plastic sheet and a permanent marker.

Carcasses were also subjectively evaluated by visual inspection for finishing and conformation (5-point rank) and for perirenal fat (3-point rank) according to Cezar and Sousa [27].

### 2.6. Production Costs and Economic Analysis

The analysis of production costs considered animal, veterinary costs, and feeding costs (in dollars), including forages, concentrate, medicines and mineral supplements used in managing the animals and the feed.

Feeding costs were obtained by multiplying the individual cost of each ingredient and the intake of each diet, and it was expressed as the average cost per animal for 52 days.

Labor costs per animal were determined considering one employee and eight-hour shifts for animal care, installations cleaning and occasional treatment of animals in the ratio of one employee taking care of 300 animals [28,29]. Minimum wage in 2018 was US$272.33 per month, totaling US$0.32/animal/day.

Finally, the revenue after animal slaughter was used, and the abovementioned costs were deducted.

The economic indexes were obtained as follows:

- B/C = liquid revenue/total cost

    In, B/C = benefit/cost ratio (>1, indicates economic viability).

- Operating liquid revenue = Total revenue − operating cost.
- Operating profit = Operating liquid revenue/Total revenue ×100

The economic analysis of the experiment was according to the method of Romão et al. [28] and Nogueira [29].

The values of buying and sale of animals per kg of live weight were:

Buying price per kg of live weight: Initial body weight multiplying US$1.54/kg of live weight.

Sale price per kg of carcass weight: Carcass weight multiplying US$4.05/kg of carcass weight.

### 2.7. Statistical Analysis

The mathematical model considered a completely randomized design with four treatments and seven animals per treatment:

$$\gamma i = \mu + \tau i + \epsilon i \tag{1}$$

Where $\gamma i$ = value observed in the plot that received treatment I; $\mu$ = overall mean; $\tau i$ = effect of treatment; and $\epsilon i$ = random error associated with treatment $i$.

Results were subjected to analysis of variance (ANOVA) and means compared by contrast Dunnett test, using the Statistical Analysis System [30], at 5% of significance.

## 3. Results

Dry matter intake was lower for the animals fed cactus pear as the only forage source and wheat bran when compared to the intake of animals fed the control diet (Table 3).

**Table 3.** Intake of diet's nutrient and water of lambs fed with cactus pear as the only forage source and different levels of wheat bran.

| Intake | Wheat Bran Levels, %DM | | | | SEM | Contrast | | | p-Value |
|---|---|---|---|---|---|---|---|---|---|
| | 0 | 30 | 37 | 44 | | 0 vs. 30 | 0 vs. 37 | 0 vs. 44 | |
| DMI, g/day | 1415.91 | 986.25 | 979.36 | 926.53 | 44.93 | * | * | * | 0.001 |
| DMI, g/kg BW | 36.19 | 28.54 | 29.14 | 26.06 | 0.98 | * | * | * | 0.001 |
| CPI, g/day | 177.93 | 122.81 | 126.27 | 108.85 | 5.89 | * | * | * | 0.001 |
| OMI, g/day | 1288.53 | 901.48 | 895.18 | 847.66 | 40.98 | * | * | * | 0.001 |
| NDFIcp, g/day | 469.74 | 243.02 | 263.25 | 260.60 | 14.43 | * | * | * | 0.001 |
| NDFIcp, g/kg BW [†] | 12.00 | 7.02 | 7.83 | 7.33 | 0.33 | * | * | * | 0.001 |
| EEI, g/day | 38.63 | 37.54 | 36.91 | 36.89 | 1.39 | ns | ns | ns | 0.115 |
| NFCI, g/day | 602.23 | 498.10 | 468.74 | 444.32 | 19.92 | * | * | * | 0.001 |
| TDNI, g/day | 1012.15 | 726.74 | 693.76 | 636.76 | 27.84 | * | * | * | 0.001 |
| Water, L/day | 1.33 | 2.07 | 1.92 | 1.70 | 0.16 | ns | ns | ns | 0.408 |

SEM—Standard error of the mean; 0% WhB was Control = buffel grass hay, cactus pear and concentrate; 30% WhB = cactus pear, 30% of wheat bran on a DM basis and concentrate; 37% WhB = cactus pear, 37% of wheat bran on a DM basis and concentrate; 44% WhB = cactus pear, 44% of wheat bran on a DM basis and concentrate; [†] BW = body weight. Dry matter intake (DMI), crude protein intake (CPI), organic matter intake (OMI), neutral detergent fiber intake (NDFI), ether extract intake (EEI), non-fibrous carbohydrate intake (NFCI), total digestible nutrient intake (TDNI); * = Statistically significant.

Similar results ($p = 0.001$) were seen for CPI, OMI, NDFI, NFCI, and TDNI (Table 3). On the other hand, the mean EEI was 37.49 g/day considering all treatments and it was not affected ($p = 0.115$) by the reduction in DMI. Voluntary water intake was not affected by the different diets ($p = 0.408$, Table 3).

The digestibility coefficients of DM were not different between the control diet and the diet with cactus pear and 30% of wheat bran (Table 4). There was also no difference between the digestibility of the organic matter of the control diet and the diets with 30 and 37% of the wheat bran and cactus pear. Nevertheless, the digestibility of DM of the control diet (731.3 g/kg) was higher than the diets with cactus pear as the only forage source and containing 37% (703 g/kg) and 44% (684.8 g/kg) of wheat bran ($p = 0.002$).

**Table 4.** In situ coefficient of digestibility of diets nutrients of lambs fed diets with cactus pear as the only forage source and levels of wheat bran.

| Digestibility, g kg$^{-1}$ | Wheat Bran Levels % DM | | | | SEM | Contrasts | | | *p*-Value |
|---|---|---|---|---|---|---|---|---|---|
| | 0 | 30 | 37 | 44 | | 0 vs. 30 | 0 vs. 37 | 0 vs. 44 | |
| DMD | 731.33 | 728.06 | 703.79 | 684.38 | 8.89 | ns | * | * | 0.002 |
| CPD | 761.27 | 699.03 | 711.56 | 651.50 | 12.83 | * | * | * | 0.001 |
| OMD | 759.05 | 754.51 | 733.04 | 715.43 | 8.19 | ns | ns | * | 0.003 |
| NDFD | 556.66 | 477.26 | 462.98 | 428.79 | 22.60 | * | * | * | 0.003 |
| EED | 865.30 | 891.02 | 841.57 | 837.24 | 10.50 | ns | ns | ns | 0.509 |
| NFCD | 899.07 | 900.67 | 881.07 | 881.17 | 9.75 | ns | ns | ns | 0.459 |
| TDN | 737.07 | 716.09 | 709.76 | 688.39 | 7.81 | ns | ns | ns | 0.707 |

SEM—Standard error of the mean; 0% WhB was Control = buffel grass hay, cactus pear and concentrate; 30% WhB = cactus pear, 30% of wheat bran on a DM basis and concentrate; 37% WhB = cactus pear, 37% of wheat bran on a DM basis and concentrate; 44% WhB = cactus pear, 44% of wheat bran on a DM basis and concentrate. Coefficient of digestibility of the dry matter (DMD), protein (CPD), organic matter (OMD), neutral detergent fiber (NDFD), ether extract (EED) and non-fibrous carbohydrates (NFCD), and total digestible nutrients (TDN); * = Statistically significant.

The digestibility coefficients of ether extract, non-fibrous carbohydrates and total digestible nutrients were not affected by diets, and mean values were 858.78 g kg$^{-1}$, 890.50 g kg$^{-1}$, and 712.83 g kg$^{-1}$, respectively (Table 4).

Crude protein ($p = 0.001$) and neutral detergent fiber ($p = 0.003$) digestibilities were different between diets. The CP digestibility of the control diet (761.2 g kg$^{-1}$) was higher than the digestibility of the diets with cactus pear as the only forage source and 30% (699.3 g kg$^{-1}$), 37% (711.6 g kg$^{-1}$), or 44% (651.0 g kg$^{-1}$) of wheat bran. NDF digestibility was also higher on the control diet (556.6 g kg$^{-1}$) when compared to the diets with 30% (477.6 g kg$^{-1}$), 37% (462.8 g kg$^{-1}$), or 44% (428.9 g kg$^{-1}$) of wheat bran.

The diets affected FBW, ADG, and TWG (Table 5). Animals fed cactus pear as the only forage source and wheat bran had lower ADG and TWG than control animals, with averages of 230 g kg$^{-1}$ (30% and 44%) and 220 g kg$^{-1}$ (37%) for ADG and averages of 12.10 kg (30%), 11.36 kg (37%), and 12.09 kg (44%) for TWG. Final body weight was similar between control and 44% WhB animals, whereas control animals had higher FBW than 30% WhB and 37% WhB.

Feed conversion ($p = 0.703$) and feed efficiency ($p = 0.915$) were not affected by diets and the average values were 5.81 kg DMI/kg BW and 178.02 g of ADG/kg DMI, respectively.

There was also no effect of diets on EBW ($p = 0.137$), HCW ($p = 0.299$), CCW ($p = 0.286$), CL ($p = 0.226$) and loin eye area ($p = 0.777$) (Table 6), with averages of 29.89 kg; 16.85 kg; 16.48 kg; 56.60%; 2.17 and 12.23 cm$^2$, respectively.

**Table 5.** Performance of lambs fed diets with cactus pear as the only forage source and levels of wheat bran.

| Itens | Wheat Bran Levels, % DM | | | | SEM | Contrasts | | | *p*-Value |
|---|---|---|---|---|---|---|---|---|---|
| | 0 | 30 | 37 | 44 | | 0 vs. 30 | 0 vs. 37 | 0 vs. 44 | |
| DMI, g/day | 1415.91 | 986.25 | 979.36 | 926.53 | 0.05 | * | * | * | 0.001 |
| IBW, kg | 22.09 | 22.39 | 22.25 | 23.62 | 0.91 | ns | ns | ns | 0.617 |
| FBW, kg | 39.29 | 34.50 | 33.68 | 35.72 | 1.29 | * | * | ns | 0.023 |
| ADG, kg/day | 0.33 | 0.23 | 0.22 | 0.23 | 0.01 | * | * | * | 0.001 |
| TWG, kg | 17.20 | 12.10 | 11.36 | 12.09 | 0.80 | * | * | * | 0.001 |
| Feed:BW, kg DMI/kg BW | 5.45 | 6.14 | 6.02 | 5.65 | 0.48 | ns | ns | ns | 0.703 |
| Feed efficiency, kg BW/kg DMI | 0.185 | 0.173 | 0.175 | 0.178 | 0.17 | ns | ns | ns | 0.915 |

SEM—Standard error of the mean; 0% WhB was Control = buffel grass hay, cactus pear and concentrate; 30% WhB = cactus pear, 30% of wheat bran on a DM basis and concentrate supplement; 37% WhB = cactus pear, 37% of wheat bran on a DM basis and concentrate; 44% WhB = cactus pear, 44% of wheat bran on a DM basis and concentrate; * = Statistically significant.

**Table 6.** Carcass characteristics of lambs fed diets with cactus pear as the only forage source and levels of wheat bran.

| Traits | Wheat Bran Levels, % DM | | | | SEM | Contrasts | | | *p*-Value |
|---|---|---|---|---|---|---|---|---|---|
| | 0 | 30 | 37 | 44 | | 0 vs. 30 | 0 vs. 37 | 0 vs. 44 | |
| IBW, kg | 22.09 | 22.39 | 22.26 | 23.62 | 0.915 | ns | ns | ns | 0.617 |
| FBW, kg | 39.28 | 34.50 | 33.62 | 35.71 | 1.594 | * | * | ns | 0.023 |
| Post-fast weight | 37.83 | 32.95 | 31.97 | 33.97 | 1.429 | * | * | ns | 0.017 |
| EBW, kg | 31.94 | 29.10 | 29.11 | 29.39 | 0.584 | ns | ns | ns | 0.137 |
| HCW, kg | 17.96 | 16.58 | 15.89 | 16.96 | 0.772 | ns | ns | ns | 0.299 |
| CCW, kg | 17.64 | 16.25 | 15.54 | 16.50 | 0.764 | ns | ns | ns | 0.286 |
| HCY, g/100g | 47.48 | 50.76 | 50.11 | 49.45 | 0.791 | * | * | ns | 0.045 |
| CCY, g/100g | 46.65 | 49.72 | 48.99 | 48.48 | 0.792 | * | * | ns | 0.066 |
| Chilling loss, kg | 1.74 | 2.05 | 2.47 | 2.42 | 0.141 | ns | ns | ns | 0.226 |
| Loin eye área, cm$^2$ | 12.63 | 12.44 | 11.58 | 12.48 | 0.390 | ns | ns | ns | 0.777 |
| Weight of non-carcass components | | | | | | | | | |
| Full GIT, kg | 9.29 | 6.60 | 6.59 | 7.63 | 0.243 | ns | ns | ns | 0.632 |
| Empty GIT, kg | 3.39 | 3.05 | 3.03 | 3.04 | 0.142 | ns | ns | ns | 0.202 |

SEM—Standard error of the mean; 0% WhB was Control = buffel grass hay, cactus pear and concentrate; 30% WhB = cactus pear, 30% of wheat bran on a DM basis and concentrate; 37% WhB = cactus pear, 37% of wheat bran on a DM basis and concentrate; 44% WhB = cactus pear, 44% of wheat bran on a DM basis and concentrate; * = Statistically significant.

Similar to FBW results, post-fast weight loss, HCY and CCY were not different between control and 44% WhB animals, and higher in control lambs when compared with 30% WhB and 37% WhB animals. Carcass measurements were not affected by experimental diets (Table 7).

**Table 7.** Morphological measurements (cm) of the carcass of lambs fed diets with cactus pear as the only forage source and levels of wheat bran.

| Traits, cm | Wheat Bran Levels, % DM | | | | SEM | Contrasts | | | *p*-Value |
|---|---|---|---|---|---|---|---|---|---|
| | 0 | 30 | 37 | 44 | | 0 vs. 30 | 0 vs. 37 | 0 vs. 44 | |
| External length | 60.43 | 58.83 | 57.57 | 58.57 | 1.28 | ns | ns | ns | 0.222 |
| Internal length | 62.14 | 59.41 | 59.17 | 59.36 | 0.98 | ns | ns | ns | 0.332 |
| Leg length | 37.64 | 36.41 | 36.79 | 37.29 | 0.68 | ns | ns | ns | 0.613 |
| Hind width | 21.29 | 20.33 | 19.57 | 20.43 | 0.69 | ns | ns | ns | 0.382 |
| Thorax width | 16.14 | 15.00 | 15.28 | 15.71 | 0.42 | ns | ns | ns | 0.276 |
| Thorax perimeter | 70.86 | 68.67 | 68.43 | 69.86 | 1.13 | ns | ns | ns | 0.406 |
| Hind perimeter | 58.43 | 58.33 | 56.43 | 59.00 | 0.93 | ns | ns | ns | 0.246 |
| External thorax depth | 25.57 | 25.50 | 25.14 | 25.43 | 0.61 | ns | ns | ns | 0.960 |
| Internal thorax depth | 27.29 | 26.50 | 26.35 | 27.07 | 0.42 | ns | ns | ns | 0.363 |

SEM—Standard error of the mean; 0% WhB was Control = buffel grass hay, cactus pear and concentrate; 30% WhB = cactus pear, 30% of wheat bran on a DM basis and concentrate; 37% WhB = cactus pear, 37% of wheat bran on a DM basis and concentrate; 44% WhB = cactus pear, 44% of wheat bran on a DM basis and concentrate.

Among the commercial cuts, only the weight of ribs and loin of lambs were affected by the experimental diets (Table 8).

**Table 8.** Weight and yield of commercial cuts of lambs fed diets with cactus pear as the only for-age source and levels of wheat bran.

| Weight | Wheat Bran Levels, %DM | | | | SEM | Contrast | | | *p*-Value |
|---|---|---|---|---|---|---|---|---|---|
| | **0** | **30** | **37** | **44** | | **0 vs. 30** | **0 vs. 37** | **0 vs. 44** | |
| Neck, kg | 1.15 | 1.06 | 1.14 | 1.15 | 0.07 | ns | ns | ns | 0.833 |
| Shoulder, kg | 1.59 | 1.53 | 1.46 | 1.57 | 0.07 | ns | ns | ns | 0.610 |
| Rib, kg | 2.63 | 2.30 | 2.23 | 2.25 | 0.13 | ns | * | * | 0.127 |
| Loin, kg | 1.04 | 0.94 | 0.87 | 0.96 | 0.05 | ns | * | * | 0.145 |
| Leg, kg | 2.78 | 2.57 | 2.47 | 2.71 | 0.12 | ns | ns | ns | 0.325 |
| Yield of commercial cuts (%) | | | | | | | | | |
| Neck | 6.51 | 6.52 | 6.63 | 6.95 | 0.19 | ns | ns | ns | 0.194 |
| Shoulder | 18.02 | 18.83 | 18.79 | 19.03 | 0.10 | * | * | * | 0.069 |
| Rib | 29.82 | 28.31 | 28.70 | 27.27 | 0.24 | * | * | * | 0.051 |
| Loin | 11.79 | 11.57 | 11.20 | 11.64 | 0.96 | ns | ns | ns | 0.700 |
| Leg | 31.51 | 31.63 | 31.79 | 32.85 | 0.16 | ns | ns | ns | 0.601 |

SEM—Standard error of the mean; 0% WhB was Control = buffel grass hay, cactus pear and concentrate; 30% WhB = cactus pear, 30% of wheat bran on a DM basis and concentrate; 37% WhB = cactus pear, 37% of wheat bran on a DM basis and concentrate; 44% WhB = cactus pear, 44% of wheat bran on a DM basis and concentrate; * = Statistically significant.

Nevertheless, the shoulder and ribs of lambs fed diets with cactus pear and 37% and 44% of wheat bran had lower weight and yield than those of animals fed the control diets. These cuts had also lower yield in animals fed 30% of wheat bran and cactus pear.

The feeding cost of the control diet/animal (US$3.37) was higher than the cost of the diet/animal with cactus pear as the only forage source and 44% of wheat bran (US$1.81, Table 9). Nevertheless, the gross revenue and total revenue per animal unit was higher for the control diet and lower in the diet of animals fed cactus pear and 37% of wheat bran.

**Table 9.** Production costs of lambs fed diets with cactus pear as the only forage source and levels of wheat bran.

| Costs | Wheat Bran Levels, % DM | | | |
|---|---|---|---|---|
| | **0** | **30** | **37** | **44** |
| Animals Total (und) | 7 | 7 | 7 | 7 |
| Cost buy animal (animal/US$) | 34.17 | 34.40 | 34.42 | 36.54 |
| Feeding DMI g/animal/day | 1415.91 | 986.25 | 979.36 | 926.53 |
| Cost Diet US$/kg DM/day | 0.32 | 0.27 | 0.26 | 0.26 |
| Cost US$/kg DM/animal/day | 0.45 | 0.27 | 0.26 | 0.24 |
| Cost Diet/animal (US$) | 3.37 | 1.98 | 1.94 | 1.81 |
| Health (US$/animal) | 0.28 | 0.28 | 0.28 | 0.28 |
| Labor cost (US$/animal/day) | 0.33 | 0.33 | 0.33 | 0.33 |
| Total revenue per animal (US$/und) | 78.30 | 70.89 | 69.89 | 74.81 |
| Cost group | | | | |
| Animal buy Cost (US$) | 239.19 | 240.80 | 240.94 | 255.78 |
| Cost Diet | 175.24 | 102.96 | 100.88 | 94.12 |
| Health (US$) | 1.99 | 1.99 | 1.99 | 1.99 |
| Labor cost (US$) | 118.03 | 118.03 | 118.03 | 118.03 |
| Total operating cost (US$) | 295.26 | 222.98 | 220.9 | 214.14 |
| Total revenue (US$ total) | 548.11 | 496.22 | 489.24 | 523.69 |

SEM—Standard error of the mean; 0% WhB was Control = buffel grass hay, cactus pear and concentrate; 30% WhB = cactus pear, 30% of wheat bran on a DM basis and concentrate; 37% WhB = cactus pear, 37% of wheat bran on a DM basis and concentrate; 44% WhB = cactus pear, 44% of wheat bran on a DM basis and concentrate.

The different wheat bran levels affected economic traits, except operating liquid revenue, as shown in Table 10.

**Table 10.** Economic traits of lambs fed diets with cactus pear as the only forage source and levels of wheat bran.

| Economic Traits Total/Group | Wheat Bran Levels % DM | | | | SEM | Contrasts | | | *p*-Value |
|---|---|---|---|---|---|---|---|---|---|
| | 0 | 30 | 37 | 44 | | 0 vs. 30 | 0 vs. 37 | 0 vs. 44 | |
| OLR [†] (US$) | 252.85 | 273.24 | 268.34 | 309.55 | 1.391 | ns | ns | ns | 0.153 |
| B/C ratio [‡] (US$) | 0.92 | 1.28 | 1.27 | 1.51 | 0.054 | * | * | * | 0.008 |
| OP [§] (%) | 4.75 | 5.60 | 5.56 | 6.00 | 0.111 | * | * | * | 0.002 |

[†] Operating liquid revenue; [‡] Cost:Benefit ratio; [§] Operating profit; SEM—Standard error of the mean; 0% WhB was Control = buffel grass hay, cactus pear and concentrate; 30% WhB = cactus pear, 30% of wheat bran on a DM basis and concentrate; 37% WhB = cactus pear, 37% of wheat bran on a DM basis and concentrate; 44% WhB = cactus pear, 44% of wheat bran on a DM basis and concentrate; * = Statistically significant.

## 4. Discussion

The quality of diet fiber may directly influence the voluntary intake, since there are interactions between energetic demands and fill capacity in ruminants [16]. Wheat bran is an ingredient with low-degradable fiber and approximately 10% of lignin, as compared to the 3%-lignin levels of soybean meal [7]. Thus, increasing dietary levels of wheat bran result in a decrease of dry matter intake (DMI), as shown in the present study and as suggested by Conceição et al. [31]. Furthermore, the high levels of non-fibrous carbohydrates of the diets containing cactus pear as the only forage source and wheat bran levels have probably contributed to the lower dry matter intake. As a consequence of lower DMI, the intakes of crude protein, neutral detergent fiber, and total digestible nutrients were also lower.

The replacement of the buffel grass hay with wheat bran in the diets resulted in lower NDF as well as lower NDFpe. NDFpe stimulate chewing and rumen motility and can promote the lower DMI in these treatments because, when the cactus is the only roughage source, due the high non-fiber carbohydrate associated to the lower NDFpe that despite promoting an increase in the microbial population, consequently increases the supply of volatile fatty acids per gram of feed, which can cause a rapid drop in pH, thus it has an inhibitory action on the intake of dry matter. Despite the above, the literature reports that the high concentration of pectin in the cactus pear is mainly responsible for preventing the sharp drop in ruminal pH, despite presenting a high rate of degradability and being analytically quantified as non-fibrous carbohydrate, as the pectin makes up the middle lamella of the plant cell wall and is a structural carbohydrate such as hemicellulose and cellulose, and due to the chemical characteristics similar to being degraded, pectin is degraded by the acidic route, resulting in the elevation of the acetate which, due to the pka, does not cause a rapid reduction in ruminal pH [32,33].

In ruminant feeding, the adequate proportion of non-fibrous carbohydrates is essential for rumen health and animal performance, with a maximum of 44% of NFC in the diets for optimal ruminal function [34–37].

The voluntary water intake was similar between treatments, probably because the amount of cactus pear was similar between the diets. Lower digestibility of crude protein may result because the lower soybean meal levels when the diets have more wheat bran level. Although Bispo et al. [35] stated that the main difference between soybean meal and wheat bran is that the wheat bran contains 21.64% of acid detergent indigestible protein (ADIP) compared with 4.11% ADIP from soybean meal.

Soares et al. [9] reported that the wheat bran particle size is smaller and less dense, resulting in lower passage rate in the rumen and it was associated to the greater complexation of cellulose with lignin in the cell wall resulting in less degradation and utilization by ruminal microorganisms. The lower DMI and digestibility generates lower FBW, ADG and TWG. According to Felix et al. [38], dry matter intake is one of the most important factors affecting performance, since it is responsible for nutrient input that is necessary to fulfil

the requirements of animals. Besides, animals fed the control diet had higher intake and retention of nitrogen that contributes to better performance [34–38].

Cactus pear used as the only forage source had no effect on feed conversion and feed efficiency of the animals. Thus, the results of feed conversion and feed efficiency showed that the lower ADG was due to lower DMI of animals fed cactus pear as the only forage, therefore, the diets were metabolically well utilized and even with the lower gain, the confinement will depend on a lower diet provided, which helps to explain the better economic results of diets with wheat bran.

The similar final body weight between control animals and those fed 44% wheat bran and cactus pear prove that the protein and energy supply provided by the diets are similar to the nutrient requirements.

Hot carcass yield was smaller in the control group due to the greater weight of the gastrointestinal tract (9.29 kg), whereas the weight was 6.94 kg in average in the other diets, since diets with greater fiber content remain during more time in the GIT and induce smaller passage rates. Hot and cold carcass yield are highly favorable, since carcass yield relates both to meat production and carcass value [39].

Diets with cactus pear as the only forage source and wheat bran provided carcass characteristics (traits) similar to the control animals, and it was not affected by diet. Similarly, the loin eye area was not different between diets. Loin eye area reflects the muscle development and the amount of meat in the carcass [40] since it is related to the total amount of muscle in the ovine carcass [40,41].

The diets had no negative effect on the morphometric measurements of the carcass, suggesting that farmers can use diets with cactus pear as the only roughage source and levels of wheat bran as a viable approach to decrease feeding costs and increase profit.

The weights and yields of the majority of commercial cuts were similar between the diets, probably because slaughter body weight was similar between treatments after confinement. Independent of breed, the proportions of almost every region of the body are usually similar when carcass weight and carcass fat proportion are similar [41–43].

Feeding costs were greater in the control animals, due to the greater cost of the buffel grass hay and the higher DMI, representing 53.71% more of the total cost with feeding when compared to animals fed the diet with cactus pear and 44% wheat bran.

The economic viability evaluation showed that the operating liquid revenue was not different between diets with cactus pear and wheat bran and the control diet with buffel grass hay and cactus pear. Nevertheless, the diet with cactus pear as the only forage source and 44% of wheat bran showed higher benefit/cost ratio (B/C ratio) comprises of the amount of capital returned to the farmer for each unit of invested capital. The analyses of these results indicate that the better ratio was 1.51 per animal, which was provided by the diet with cactus pear as the only forage source and 44% wheat bran. This value means that for each US$1.00 invested, when this diet was used in the confinement, the expected return is US$1.51 at the end of the production period.

Some slaughterhouses have stimulated carcasses with better finishes through higher selling prices [44]. Considering the higher numeric carcass yields, it seems that they had more fat finish and would be subsidized with higher selling prices, thus increasing even more the economic viability of the use of cactus pear as an only roughage source associated with a wheat bran.

The use of cactus pear as an only roughage source associated with a wheat bran was shown to be efficient for weight gain in lambs and had no effect on carcass quality and composition. Therefore, in situations when only cactus pear is available as a roughage source, it is possible associate with wheat bran to promote positive effects in economic viability.

## 5. Conclusions

The use of cactus pear as the only roughage source associated with up to 44% wheat bran is a viable alternative for the diet of confined lambs.

When the cactus pear was combined up to 37% it promoted higher yields of hot carcass and cold carcass, increasing the quality of the carcass produced.

The greater benefit:cost ratio was obtained when the cactus pear was the only source of forage in association with 44% of wheat bran.

**Author Contributions:** K.B.d.S.; J.S.d.O.; E.M.S.; J.P.d.F.R.; F.Q.C.; P.E.N.G.; A.F.d.N.S.; G.F.d.L.C.; J.M.C.N.; and J.P.A. put together the concept and design of this study and conducted the experiments. J.S.d.O.; E.M.S.; and A.d.M.Z. performed formal analysis. J.S.d.O.; E.M.S.; D.d.J.F.; A.G.V.d.O.L. reviewed the manuscript and provided resources. J.S.d.O.; E.M.S. wrote the manu-script and super-vised the study. All authors have read and agreed to the published version of the manuscript.

**Funding:** This work was supported by the Financing of Research Innovation (FINEP), as part of the Agrocap group; Research Scientific and Technological Development of Maranhão (FAPEMA), Coordination for the Improvement of Higher Education Personnel (CAPES-Brazil) and National Council for Scientific and Technological Development (CNPq-Brazil) for the fellowship grant; and the Integrated Animal Science Doctoral Program in partnership with the Federal University of Paraíba-Brazil (UFPB).

**Institutional Review Board Statement:** All procedures performed in studies involving animals were in accordance to the guidelines of the Declaration of Helsinki and approved by the Ethics Committee of Federal University of Paraiba (protocol code 8179070318 at June of 2007).

**Informed Consent Statement:** Not applicable.

**Data Availability Statement:** The data presented in this study are available on request from the corresponding author.

**Conflicts of Interest:** The authors declare that they have no conflict of interest.

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
