# Peer review of "Cactus Pear as Roughage Source Feeding Confined Lambs: Performance, Carcass Characteristics, and Economic Analysis"

_agronomy, doi:10.3390/agronomy11040625_

Round 1

Reviewer 1 Report

Manuscript ID: agronomy-1134414

Cactus Pear as Roughage Source Feeding Confined Lambs: Performance, Carcass Characteristics and Economic Analysis

General remarks

Dear Authors,

I have revised the abovementioned manuscript. The study of potential use potential of alternative forage sources and agro-industrial by-products is an extremely interesting topic and, therefore, the authors' contribution to literature is appreciated. Nevertheless, in my opinion, the manuscript needs to be carefully revised in several parts. My major concerns regard the materials and methods section, abstract, and part of the discussions. My suggestions are detailed below. Hoping to have contributed to improving the manuscript quality, I wish the authors a good job.

Specific comments

L3 (title) and L20 (abstract): please, add a comma before “and Economic Analysis”. Thanks.

L17: in my opinion, the authors should specify the hypothesis under which they used wheat bran in combination with cactus pear cladodes. The reasons can be understood in the subsequent simple summary, but they are absolutely not deducible from the abstract.

L26: in my opinion, the use of abbreviations in the abstract should be avoided. Authors are kindly requested to consult the instructions in this regard.

L30: please, replace “as the only roughage source” with “was used as the only roughage source”.

L32: please, add “of” after wheat bran as well as “the” after diet, and replace modify with “modifying”.

L33: please, add “ratio” to cost: benefit. Thanks.

L35; please, add “a” after source and comma after and energy.

L35-37: in my opinion, the sentence should be changed, as follow: However, when a cactus pear is used as the only roughage source in ruminant feeding, liquid feces, and weight loss can occur.

L37-39: in my opinion, the sentence' syntax should be improved.

L43: please, delete the comma after 44%. Thanks.

L44 (keywords”):  I think "confinement" is not an appropriate keyword. In addition, the authors did not mention the cactus pear cladodes, which is the main object of the study.

L51: animal confinement in a semi-arid environment to minimizing the consequences of forage scarcity is a common strategy, adopted to livestock farms in many areas of the world. To avoid unnecessary localism, I think it appropriate that the authors also refer to other productive contexts. From this point of view, in addition to Neto et al. 2016 (which refers to Latin America), the authors may also refer to https://doi.org/10.3390/foods9081091 and https://doi.org/10.1016/j.smallrumres.2020.106085 (which I encourage to use as references). Thanks.

L72: I notice an excess of “lambs”!

L90: since the authors rely on the forage scarcity in an arid climate, I believe it is appropriate to classify the climate under which the experiment was conducted. In this regard, Köppen' climate classification could be useful. Thanks.

L92: What breeds were crossed? This information has its own importance for interpreting the daily weight gain and the carcass yield of lambs. Among other things, compared to the age declared by the authors, the weight of the lambs seems a little low, so I think it is further necessary to know the genetic types from which the lambs derive. Thanks.

L96-97: with a view to compliance with animal welfare regulations, I believe it appropriate that the authors provide more details about the type of cage (e.g., cage size) and the daily care provided to the lambs. Thanks.

L122 (and along with the text): please, replace “leftover” with feed refusals. Thanks.

L123-126: I congratulate the authors for considering the evaporation rate.

L135-138: in my opinion, this part is redundant since is reported similarly in the apparent digestibility section.

L143: I believe a sentence cannot be started with a reference.

L145: I believe FDN means NDF.

L154: please, replace “NDFpe” with peNDF. Thanks.

L154 (and along with the manuscript): I suggest authors always indicate the author of the reference in the text (e.g., Nogueira [21]) to make the reading more fluent.

L154-159: Is the sieve to which authors refer to the Penn State Particle Separator? If yes, the authors are requested to report it in the text. Otherwise, the authors are asked to better detail the used equipment. Thanks.

L169: please, delate “laboratorial”. Thanks.

L170: authors are invited to avoid starting sentences with acronyms.

L170: in my opinion, the sentence is unclear and need to recast (e.g. “The iNDF concentrations were determined by incubation of samples of concentrates (1g) hay, feces, and leftovers (0.5g) in a non-woven-fabric bag inserted for hours in the rumen of a fistulated bovine”). Furthermore, the authors should specify how long the incubation lasted, what type of animal was used (bovine is too generic), the characteristics of diet fed to fistulated animals.

L218-220: in my opinion, veterinary costs are more appropriate. Anyhow, costs for medicines and other management costs are not' feeding costs. Finally, what tool costs were considered? Authors are invited to clarify.

L222: was expressed instead of “is expressed”.

L223-225: in my opinion, it is not clear if the cost of labor was calculated as the proportion of 300 animals per worker or based on the 8 hours dedicated to each animal mentioned above. I think that sentence needs to recast to avoid misunderstanding.

L225-226: please delete “which”. Anyhow, I have some difficulty understanding how the daily cost per head was calculated. The authors could provide some more computational notes in this regard.

L234: see suggestion referring to L154.

L234: in my opinion, the calculation method should be reported before formulas explication.

L238-239: in many parts of the World, the carcass selling price can be varied according to its conformation score. The authors are asked to clarify why this has not been considered. Thanks.

L254 (and along with the text): each table should be explanatory. Authors are requested to report all the acronyms used in the table' footnotes.

L276: calculation of the ether extract digestibility coefficient was not previously mentioned in the materials and methods section.

L382-464: tables' citations should be avoided along with the discussions since the tabulated results have already been discussed in the appropriate section.

L388 (and along with the manuscript): see suggestion referring to L154.

L399-403: from my point of view, I do not think it appropriate for the authors to discuss data not shown in the results. Have rumen pH, ammonia levels, and volatile fatty acid concentration been determined? If so, authors are encouraged to describe this in the materials and methods and report the data in the results; vice versa, I do not think it is possible to limit us only to discussing them.

L413, 417: see suggestion referring to L154.

L435-464: this manuscript part seems more like a simple recall of the results than a real discussion. Authors are invited to improve it if possible.

Author Response

Ms. Cassie Xu

Assistant Editor

Agronomy

We are pleased our paper was accepted and appreciate the attention and contributions the article received from your team of reviewers. All reviewer comments have been addressed, as indicated below and in the attached file. We carefully observed the style and form of Agronomy and revised the manuscript accordingly. After all the corrections requested, if necessary, we have committed that if necessary, we can submit the manuscript for correction by a native speaker company. The responses to the questions/comments by the reviewers are provided below, and all the changes in the manuscript are highlighted in yellow.

Sincerely yours,

Anderson de Moura Zanine

Reviewer #1

Comments for the author

Correction and answers

General comments

I have revised the abovementioned manuscript. The study of potential use potential of alternative forage sources and agro-industrial by-products is an extremely interesting topic and, therefore, the authors' contribution to literature is appreciated. Nevertheless, in my opinion, the manuscript needs to be carefully revised in several parts. My major concerns regard the materials and methods section, abstract, and part of the discussions. My suggestions are detailed below. Hoping to have contributed to improving the manuscript quality, I wish the authors a good job

We have been grateful for the attention and time dedicated to the evaluation of our manuscript. We have corrected the manuscript with your suggestions to improving quality and understanding.

Specific Comments

L3 (title) and L20 (abstract): please, add a comma before “and Economic Analysis”. Thanks

We have added as you suggested.

L17: in my opinion, the authors should specify the hypothesis under which they used wheat bran in combination with cactus pear cladodes. The reasons can be understood in the subsequent simple summary, but they are absolutely not deducible from the abstract.

We have been grateful for the attention and suggestion. However, we have understood that the hypothesis must come in the simple summary and in the introduction.

L26: in my opinion, the use of abbreviations in the abstract should be avoided. Authors are kindly requested to consult the instructions in this regard.

We have been grateful for the attention and suggestion. We have replaced the abbreviations with the words.

L30: please, replace “as the only roughage source” with “was used as the only roughage source”.

We have replaced “as the only roughage source” with “was used as the only roughage source”.

L32: please, add “of” after wheat bran as well as “the” after diet, and replace modify with “modifying”.

We have added all that your suggestion.

L33: please, add “ratio” to cost: benefit. Thanks.

We have added all that your suggestion.

L35; please, add “a” after source and comma after and energy.

We have disagreed to added “a” after source, but we have added the comma after and energy.

L35-37: in my opinion, the sentence should be changed, as follow: However, when a cactus pear is used as the only roughage source in ruminant feeding, liquid feces, and weight loss can occur.

We have modified as you suggested.

L37-39: in my opinion, the sentence' syntax should be improved.

We have modified as you suggested.

L43: please, delete the comma after 44%. Thanks.

We have modified as you suggested.

L44 (keywords”):  I think "confinement" is not an appropriate keyword. In addition, the authors did not mention the cactus pear cladodes, which is the main object of the study.

We have replaced "confinement" with cactus pear cladodes.

L51: animal confinement in a semi-arid environment to minimizing the consequences of forage scarcity is a common strategy, adopted to livestock farms in many areas of the world. To avoid unnecessary localism, I think it appropriate that the authors also refer to other productive contexts. From this point of view, in addition to Neto et al. 2016 (which refers to Latin America), the authors may also refer to https://doi.org/10.3390/foods9081091 and https://doi.org/10.1016/j.smallrumres.2020.106085 (which I encourage to use as references). Thanks.

We have added the reference (https://doi.org/10.1016/j.smallrumres.2020.106085) as you suggested.

L72: I notice an excess of “lambs”!

We have removed the excess of lambs

L90: since the authors rely on the forage scarcity in an arid climate, I believe it is appropriate to classify the climate under which the experiment was conducted. In this regard, Köppen' climate classification could be useful. Thanks.

We have added as you suggested.

L92: What breeds were crossed? This information has its own importance for interpreting the daily weight gain and the carcass yield of lambs. Among other things, compared to the age declared by the authors, the weight of the lambs seems a little low, so I think it is further necessary to know the genetic types from which the lambs derive. Thanks.

Thank you. The lambs were undefined crossbred because this daily weight gain and the carcass yield were lower than other lamb breeds.

L96-97: with a view to compliance with animal welfare regulations, I believe it appropriate that the authors provide more details about the type of cage (e.g., cage size) and the daily care provided to the lambs. Thanks.

We have added cage size and the daily care provided to the lambs.

L122 (and along with the text): please, replace “leftover” with feed refusals. Thanks.

We have replaced as you requested

L123-126: I congratulate the authors for considering the evaporation rate.

Thank you! We have understood that in the environmental conditions of our trial were necessary to measure the evaporation rate.

L135-138: in my opinion, this part is redundant since is reported similarly in the apparent digestibility section.

We have agreed however we have understood that necessary to describe all the methodologies.

L143: I believe a sentence cannot be started with a reference.

We have corrected the sentence.

L145: I believe FDN means NDF.

You are correct. We have replaced.

L154: please, replace “NDFpe” with peNDF. Thanks.

We have replaced “NDFpe” with peNDF.

L154 (and along with the manuscript): I suggest authors always indicate the author of the reference in the text (e.g., Nogueira [21]) to make the reading more fluent.

We have agreed

L154-159: Is the sieve to which authors refer to the Penn State Particle Separator? If yes, the authors are requested to report it in the text. Otherwise, the authors are asked to better detail the used equipment. Thanks.

Yes, we have corrected the reference.

L169: please, delate “laboratorial”. Thanks.

We have deleted.

L170: authors are invited to avoid starting sentences with acronyms.

We have modified as you suggested.

L170: in my opinion, the sentence is unclear and need to recast (e.g. “The iNDF concentrations were determined by incubation of samples of concentrates (1g) hay, feces, and leftovers (0.5g) in a non-woven-fabric bag inserted for hours in the rumen of a fistulated bovine”). Furthermore, the authors should specify how long the incubation lasted, what type of animal was used (bovine is too generic), the characteristics of diet fed to fistulated animals.

We have added the information’s as you suggested.

L218-220: in my opinion, veterinary costs are more appropriate. Anyhow, costs for medicines and other management costs are not' feeding costs. Finally, what tool costs were considered? Authors are invited to clarify.

We have removed the cost tools because we understand that it was causing more doubts than clarifying the methodology.

L222: was expressed instead of “is expressed”.

We have replaced “is” with “was”.

L223-225: in my opinion, it is not clear if the cost of labor was calculated as the proportion of 300 animals per worker or based on the 8 hours dedicated to each animal mentioned above. I think that sentence needs to recast to avoid misunderstanding.

We have considered that the labor business was 8 hours a day, in this labor business the employee of the account of handling 300 animals, so we have considered that the salary was divided by the 300 animals.

L225-226: please delete “which”. Anyhow, I have some difficulty understanding how the daily cost per head was calculated. The authors could provide some more computational notes in this regard.

We have deleted the which.

L234: see suggestion referring to L154.

We have corrected.

L234: in my opinion, the calculation method should be reported before formulas explication.

We have disagreed. But if in your opinion it is essential that be inverted the description to before the formula it we can change.

L238-239: in many parts of the World, the carcass selling price can be varied according to its conformation score. The authors are asked to clarify why this has not been considered. Thanks.

We have agreed that the carcass selling price can be bonus in the with higher scores, however in this trial we had sold it to a slaughterhouse that does not make this bonus in price

L254 (and along with the text): each table should be explanatory. Authors are requested to report all the acronyms used in the table' footnotes

Thank you so much.

We have added as you requested.

L276: calculation of the ether extract digestibility coefficient was not previously mentioned in the materials and methods section.

Sorry, We have added the EED in the equation, also we have added the estimated of NFCD.

L382-464: tables' citations should be avoided along with the discussions since the tabulated results have already been discussed in the appropriate section.

We have agreed and we have deleted as you suggested.

L388 (and along with the manuscript): see suggestion referring to L154.

We have corrected.

L399-403: from my point of view, I do not think it appropriate for the authors to discuss data not shown in the results. Have rumen pH, ammonia levels, and volatile fatty acid concentration been determined? If so, authors are encouraged to describe this in the materials and methods and report the data in the results; vice versa, I do not think it is possible to limit us only to discussing them.

Unfortunately, ruminal pH, ammonia levels and the concentration of volatile fatty acids have not been determined. We have included in the discussion as a form of speculation, based on physiology and ruminal fermentation with reference to the literature, the possibility of favoring the reader to include these analysis in a future trial, is also the reason for the inclusion of this speculation.

L413, 417: see suggestion referring to L154.

We have corrected.

L435-464: this manuscript part seems more like a simple recall of the results than a real discussion. Authors are invited to improve it if possible.

Reviewer 2 Report

The paper entitled “Cactus Pear as Roughage Source Feeding Confined Lambs: Performance, Carcass Characteristics and Economic Analysis” by Balduíno da Silva et al, deals with the evaluation of the effect of two types of diets on castrated lambs, one containing 0% of wheat bran and one with cactus pear as the forage source and wheat bran at three different levels. Nutrient digestibility, feed intake, animal performance, carcass characteristics and economic analysis were evaluated by the Authors. The research topic is of great importance and growing relevance, in order to reduce costs and improve the productivity of livestock farms with attention to environmental protection and the use of less environmental impact techniques also in small ruminants’ farms.

The introduction describes the aims of the research and provides a well referenced knowledge background. The experimental design is adequate, providing good statistical analysis and proper exposition of the results. The paper is overall well written and easily accessible for the reader. Some English improvements are needed.

Some part of the text must be improved, such the ones specified in the comments below:

Line 47: Semiarid regions of which Country/Continent? Europe? South America? Africa? Please provide some details about the area in the introduction rather than only in the MM section.

Line 92: “It was used 28 male…” please check the English language of this phrase.

Line 160: check “digestibilityof”.

Lines 258-259: all the parameters have a P=0.001, so Authors should shorten this sentence, eg from “Similar results were seen for CPI (P=0.001), OMI (P=0.001), NDFI (P=0.001), NFCI (P=0.001) e TDNI (P=0.001) (Table 3).” to “Similar results (P=0.001) were seen for CPI, OMI, NDFI, NFCI e TDNI (Table 3).” Furthermore, the letter “e” in front of TDNI is intended to be a “and”?

Line 313: check the Table for “Ns” and “ns”.

Line 433: check for typos.

Lines 458-459: Authors use “US$ 1.00” with a space between “$” and ”1”. This is different from what was done in other part of the text (eg line 365). Please check and uniform.

Line 465: Conclusions are poor. Please consider extending it.

Lines 483-and on: References need to be rewritten. Lot of references in the top of the list are not in the proper order. Assuming that the references are correct (but it’s not possible to be sure) it’s almost impossible to properly associate them in the text. It’s mandatory to provide a good reference list to enrich the paper.

Author Response

Ms. Cassie Xu
Assistant Editor
Agronomy
We are pleased our paper was accepted and appreciate the attention and contributions the article received from your team of reviewers. All reviewer comments have been addressed, as indicated below and in the attached file. We carefully observed the style and form of Agronomy and revised the manuscript accordingly. After all the corrections requested, if necessary, we have committed that if necessary, we can submit the manuscript for correction by a native speaker company. The responses to the questions/comments by the reviewers are provided below, and all the changes in the manuscript are highlighted in yellow.
Sincerely yours,
Anderson de Moura Zanine

Reviewer #2

Comments for the author

Correction and answers

The paper entitled “Cactus Pear as Roughage Source Feeding Confined Lambs: Performance, Carcass Characteristics and Economic Analysis” by Balduíno da Silva et al, deals with the evaluation of the effect of two types of diets on castrated lambs, one containing 0% of wheat bran and one with cactus pear as the forage source and wheat bran at three different levels. Nutrient digestibility, feed intake, animal performance, carcass characteristics and economic analysis were evaluated by the Authors. The research topic is of great importance and growing relevance, in order to reduce costs and improve the productivity of livestock farms with attention to environmental protection and the use of less environmental impact techniques also in small ruminants’ farms.

We have been grateful for the attention and time dedicated to the evaluation of our manuscript. We have corrected the manuscript with your suggestions to improving quality and understanding.

Introduction

The introduction describes the aims of the research and provides a well referenced knowledge background. The experimental design is adequate, providing good statistical analysis and proper exposition of the results. The paper is overall well written and easily accessible for the reader. Some English improvements are needed.

We have corrected the manuscript with your suggestions to improving quality and understanding.

Specific comments

Line 47: Semiarid regions of which Country/Continent? Europe? South America? Africa? Please provide some details about the area in the introduction rather than only in the MM section.

We have added.

Line 92: “It was used 28 male…” please check the English language of this phrase.

We have corrected.

Line 160: check “digestibilityof”.

We have corrected

Lines 258-259: all the parameters have a P=0.001, so Authors should shorten this sentence, eg from “Similar results were seen for CPI (P=0.001), OMI (P=0.001), NDFI (P=0.001), NFCI (P=0.001) e TDNI (P=0.001) (Table 3).” to “Similar results (P=0.001) were seen for CPI, OMI, NDFI, NFCI e TDNI (Table 3).” Furthermore, the letter “e” in front of TDNI is intended to be a “and”?

We have replaced “Similar results were seen for CPI (P=0.001), OMI (P=0.001), NDFI (P=0.001), NFCI (P=0.001) e TDNI (P=0.001) (Table 3).” with “Similar results (P=0.001) were seen for CPI, OMI, NDFI, NFCI e TDNI (Table 3).”

Yes, you are correct. We have corrected. Thank you

Line 313: check the Table for “Ns” and “ns”.

We have corrected. Thank you

Line 433: check for typos.

We have corrected.

Lines 458-459: Authors use “US$ 1.00” with a space between “$” and ”1”. This is different from what was done in other part of the text (eg line 365). Please check and uniform.

We have uniformed as you suggested.

Line 465: Conclusions are poor. Please consider extending it.

We have extended the conclusion as you requested.

Lines 483-and on: References need to be rewritten. Lot of references in the top of the list are not in the proper order. Assuming that the references are correct (but it’s not possible to be sure) it’s almost impossible to properly associate them in the text. It’s mandatory to provide a good reference list to enrich the paper.

We have corrected the References

Round 2

Reviewer 1 Report

Dear authors,
I have re-examined the manuscript and, in light of the changes made, I believe it is suitable for publication. However, I am perplexed about what the authors say about the sentence on lines 385-388 (that is: "On the other hand, NDFIap did not affect the ruminal parameters of the animals in the present study such as pH (6.34), ammoniacal nitrogen levels (N-NH, 13.43 mM), and volatile fatty acids (acetic acid = 22.07 mmol L−1, propionic acid= 2.32 mmol L−1  and butyric acid = 1.41 mmol L−1), as reported elsewhere [32]"). I fully understand the willingness of the authors to discuss their results in a speculative way but, otherwise, I cannot understand the obstinacy to leverage unobserved results. So, it's okay for the authors to speculate, but not in the form it was done, i.e. by referring to data not presented. It would be advisable, in my opinion, that authors' reporting the results of other studies.
As a minor detail, I suggest changing "fontain" to "fountain" (L103). Greetings

Author Response

Ms. Cassie Xu

Assistant Editor

Agronomy

We are pleased our paper was accepted and appreciate the attention and contributions the article received from your team of reviewers. All reviewer comments have been addressed, as indicated below and in the attached file. We carefully observed the style and form of Agronomy and revised the manuscript accordingly. After all the corrections requested, if necessary, we have committed that if necessary, we can submit the manuscript for correction by a native speaker company. The responses to the questions/comments by the reviewers are provided below, and all the changes in the manuscript are highlighted in yellow.

Sincerely yours,

Anderson de Moura Zanine

Comments for the author

Correction and answers

General comments

Dear authors,

I have re-examined the manuscript and, in light of the changes made, I believe it is suitable for publication. However, I am perplexed about what the authors say about the sentence on lines 385-388 (that is: "On the other hand, NDFIap did not affect the ruminal parameters of the animals in the present study such as pH (6.34), ammoniacal nitrogen levels (N-NH, 13.43 mM), and volatile fatty acids (acetic acid = 22.07 mmol L−1, propionic acid= 2.32 mmol L−1  and butyric acid = 1.41 mmol L−1), as reported elsewhere [32]"). I fully understand the willingness of the authors to discuss their results in a speculative way but, otherwise, I cannot understand the obstinacy to leverage unobserved results. So, it's okay for the authors to speculate, but not in the form it was done, i.e. by referring to data not presented. It would be advisable, in my opinion, that authors' reporting the results of other studies.

We have been grateful for the attention and time dedicated to the evaluation and improved of our manuscript.

We have changed the paragraph, and added other references.

As a minor detail, I suggest changing "fontain" to "fountain" (L103). Greetings

We have replaced "fontain" with "fountain.
